# Preservation of scRNA-Seq Libraries Using Existing Inactivation Protocols

**DOI:** 10.3390/pathogens13020167

**Published:** 2024-02-13

**Authors:** Gail L. Sturdevant, Kimberly D. Meade-White, Sonja M. Best, Emily Speranza

**Affiliations:** 1Innate Immunity and Pathogenesis Section, Laboratory of Neurological Infections and Immunity, National Institute of Allergy and Infectious Diseases, National Institutes of Health, Hamilton, MT 59840, USA; sturdevantg@niaid.nih.gov (G.L.S.); sbest@niaid.nih.gov (S.M.B.); 2Disease Modeling and Transmission Section, Laboratory of Virology, National Institute of Allergy and Infectious Diseases, National Institutes of Health, Hamilton, MT 59840, USA; kmeade-white@niaid.nih.gov; 3Florida Research and Innovation Center, Cleveland Clinic Lerner Research Institute, Port Saint Lucie, FL 34987, USA

**Keywords:** single-cell RNA sequencing, biosafety level 4, sample inactivation

## Abstract

Single-cell RNA sequencing has soared in popularity in recent years. The ability to deeply profile the states of individual cells during the course of disease or infection has helped to expand our knowledge of coordinated responses. However, significant challenges arise when performing this analysis in high containment settings such as biosafety level 3 (BSL-3), BSL-3+ and BSL-4. Working in containment is necessary for many important pathogens, such as Ebola virus, Marburg virus, Lassa virus, Nipah and Hendra viruses. Since standard operating procedures (SOPs) for inactivation are extensive and may compromise sample integrity, we tested whether the removal of single-cell sequencing libraries from containment laboratories using existing inactivation protocols for nucleic acid extraction (Trizol, RLT buffer, or AVL buffer) was feasible. We have demonstrated that the inactivation does not affect sample quality and can work with existing methods for inactivation.

## 1. Introduction

The study of emerging infectious diseases is greatly aided by the availability of single-cell RNA sequencing technologies. With the power to deeply profile individual cellular responses to infection, we can now generate a deeper understanding of how the host response to infections is coordinated [1]. One challenge faced when working with many pathogens of high public health impact is the need for work in high containment laboratories under the biosafety level 3 and biosafety level 4 (BSL-4) conditions [2]. Work in these spaces is often limited and introduction of new technologies often requires extensive testing for the inactivation of pathogens before the sample can be removed.

Many pathogens that are studied in high containment are considered to have high pandemic potential [3], necessitating a better understanding of how the host response is coordinated. These include negative sense RNA viruses like Ebola virus, Marburg virus, and Lassa virus, as well as positive sense RNA viruses such as Kyasanur forest disease virus and related flaviviruses. Many of these pathogens fall under the regulation of the Division of Regulatory Science and Compliance (DRSC) [4]. According to DRSC, protocols for each viral family must be validated in house to ensure that it is effective at killing the pathogen [5]. For single-cell RNA sequencing studies, this approach can become cost-prohibitive.

To create a method for performing single-cell sequencing from samples generated in high containment laboratories, we have outlined a method for sample inactivation of samples infected with negative stranded RNA viruses using reagents and protocols that are commonly already approved for sample inactivation such as Trizol, RLT, and AVL buffers. We have demonstrated that this simple method of sample prep preserves the sample quality, allowing for downstream analysis.

## 2. Materials and Methods

### 2.1. Single-Cell Sample Prep

Single-cell suspensions were collected from either A549 cells or from mouse cervical lymph nodes. For A549 cells, cells were grown in Dulbecco’s MEM (DMEM) (Gibco/Thermo Fisher Scientific, Waltham, MA, USA) supplemented with 10% fetal bovine serum (Gibco/Thermo Fisher Scientific, Waltham, MA, USA). The cells were washed in phosphate-buffered saline (PBS) (Gibco/Thermo Fisher Scientific, Waltham, MA, USA), then made into a single-cell suspension using trypsin. The cells were then processed according to the 10X Genomics (10X Genomics, Pleasanton, CA, USA) protocol for preparation for use in the Chromium Controller. 

For the lymph node samples, 6–8-week-old C57Bl/6J cervical lymph nodes were collected and aggerated into a single tube. Each condition contained 6 animals, with 3 males and 3 females mixed together. The lymph nodes were placed into RPMI media (Gibco/Thermo Fisher Scientific, Waltham, MA, USA) with 2% heat-inactivated FBS and 10 mM EDTA (Gibco/Thermo Fisher Scientific, Waltham, MA, USA). Samples were manually mashed and left to incubate for 20 min at 37 °C in 100 ug/mL liberase (Fisher Scientific, Pittsburgh, PA, USA) and DNase I 20 ug/mL (Sigma-Aldrich, St. Louis, MO, USA). After incubation, 20 μL of 0.5 M EDTA was added to the samples and they were mixed for 5 min. After mixing, samples were washed through a 100 mM sized filter (BD Biosciences, Franklin Lakes, NJ, USA) and washed with 4 mL of PBS with EDTA and 2% heat-inactivated FBS. The cells were then spun for 5 min at 1200 rpm and resuspended in 50 μL of PBS media. Red blood cell lysis was performed by adding 1 mL of ACK lysis buffer (Gibco/Thermo Fisher Scientific, Waltham, MA, USA) for 4 min, the supernatant was removed, and cells were resuspended in 1 mL of PBS. Samples were then spun for 3 min at 1200 rpm and resuspended in 1 mL of PBS with EDTA and heat-inactivated FBS. Cells were counted and tested for viability. For lymphocyte removal, cells were stained for 10 min with CD90.1 and CD19 MACS biotin antibodies (Miltenyi Biotec, Gaithersburg, MD, USA). After incubation, the flow through from the MACS columns (Miltenyi Biotec, Gaithersburg, MD, USA) was collected, which contained all myeloid and stromal cells from the lymph nodes. The cells were counted on the countess for concentration and viability. They were then washed twice in DPBS supplemented with 0.1% Miltenyi BSA (Miltenyi Biotec, Gaithersburg, MD, USA) and brought to the correct concentration for the 10X Genomics protocol for single-cell sequencing.

### 2.2. Single-Cell Library Prep

Once single-cell suspensions were generated, the samples were loaded onto the Chromium chips following the 10X Genomics protocol for single-cell 3′ kits, version 3 (10X Genomics, Pleasanton, CA, USA). The 10X protocol was followed for cDNA generation, cDNA amplification, and final library generation.

### 2.3. Sample Inactivation and Re-Extraction

After the final library was generated, samples were inactivated using known inactivation protocols [6,7,8]. For the AVL method (Qiagen, Germantown, MD, USA), the final library volume was brought to 140 μL using sterile water and this was combined with 560 μL of AVL buffer. The samples were left to incubate for a minimum of 10 min at room temperature, then combined with 560 μL 200 proof ethanol and incubated another 10 min. For the RLT method (Qiagen, Germantown, MD, USA), 600 μL of RLT buffer with added β-mercaptoethanol was mixed with 140 μL of the final library. The final library was brought up to 140 μL using water. The samples were left to incubate for a minimum of 10 min at room temperature, then combined with 600 μL 70% ethanol and incubated another 10 min. For Trizol, a ratio of 1:4 of the sample to Trizol was used. Trizol was added directly to the final library volume. This was left to incubate for a minimum of 10 min at room temperature. Each sample type was then transferred to clean cryovials and fully submerged in approved disinfectant during removal from the maximum containment BSL-4 laboratory. The subsequent steps were performed at biosafety level 2.

For re-extraction from the AVL or RLT buffer, samples were processed directly using the DNA/RNA AllPrep spin columns (Qiagen). The manufacture’s protocol was followed for DNA extraction. The final libraries were eluted in 30 μL of buffer EB. The entire process of re-extraction takes about 5 min. For re-extraction from Trizol, the sample can be run following the AllPrep protocol [9]. After running the extraction, an additional sample clean-up with beads is recommended to remove any remaining Trizol reagent. To achieve this, we followed the 10X protocol for the final library bead cleanup after amplification.

After re-extraction, samples were run on the Bioanalyzer High Sensitivity DNA chip (Agilent, Santa Clara, CA, USA) to determine the average fragment length and sample concentration.

### 2.4. Single-Cell Library Data Processing

Single-cell libraries from the mouse lymph nodes were sequenced following the recommendations from 10X genomics for cycles on the Illumina NextSeq (Illumina, San Diego, CA, USA). After sequencing, samples were processed through the cellRanger pipeline (v6.0.1) from 10X Genomics for deconvolution and alignment to the mouse genome. Following alignment, matrix files were read into R (v4.2.1) using Seurat version 4 [10]. Cell IDs were determined using the method described in [11,12,13] using the Immgen Database [14] as a back end. Analysis was performed in Seurat to integrate the samples, create UMAP and PCA projections, and look at differences between conditions. To remove the effects of different abilities for the Milently beads to remove T and B cells on a per-sample basis, lymphocytes were excluded from the analysis. Additionally, samples were filtered for dead/dying cells as well as doublet populations. Gene markers for each cell type can be seen in Figure 1 in addition to a UMAP with the annotated cell types.

### 2.5. Statistic

The type of test used in the analysis is presented in the figure legend. All *p*-values are presented as numbers on the figures. For pairwise comparisons, a non-parametric Mann–Whitney test was used. Graphs and statistics were generated in Prism 10 (version 10.1.1).

## 3. Results

### 3.1. Sample Stablilty and Inactivation Procedure

The generation of sequencing libraries for single-cell RNA studies involves multiple steps. These include generation of a highly viable single cell suspension, barcoding of individual transcripts at the single-cell level, generation of single-stranded cDNA, amplification of cDNA, fragmentation of samples down to size, ligation of sequencing primers, and final amplification of the library. Through all of these steps, the component that takes the most amount of hands-on time is the generation of the single-cell suspension before the samples are run through the barcoding step (Figure 2). To determine an optimal stage to test the inactivation procedures, we looked for when the sample is at the most stable stage (Figure 2) and determined it was when the final library was present. At this stage, the sample is in the form of short-fragment double-stranded DNA. Taking samples to the final library stage has many advantages. First, the sample is highly amplified, meaning small losses will not have a large effect on the data quality. Second, all single-cell sequencing platforms including 10X, SPLiT-Seq [15], PiP-Seq [16], Microwell Seq [17], and Seq-well [18] will contain a final library that is a similar composition of short fragments of amplified DNA libraries with Illumina adaptors on the ends. This creates consistency across any technology that is adopted in the laboratory. Third, the final libraries are very stable and are most likely to handle inactivation with chemicals.

To test the stability of the final library against multiple commonly approved methods of inaction for nucleic acid removal from high containment, we used the 10X genomics NextGEM technology. Various types of samples (lymph nodes and A549 cells) were brought to a single-cell suspension and run through the 10X protocol for library preparation. Some samples were subjected to inactivation and re-extraction and others were analyzed as is without re-extraction. The inactivation that was tested included the AVL or RLT buffer from Qiagen and Trizol reagent from Invitrogen, which have all been demonstrated to inactivate enveloped viruses [19,20]. Following library prep, samples were inactivated according to the approved SOP for nucleic acid extraction (see Section 2). Once inactivated, the AVL or RLT samples were processed through the Qiagen AllPrep columns according to the manufacturer’s protocol for DNA extraction. The Trizol samples were processed through the AllPrep columns with additional bead cleanup after elution to remove any excess Trizol reagent. The samples were then analyzed using the bioanalyzer to determine before and after inactivation concentrations. There was no difference in the fragment size that was seen after the re-extraction following inactivation (*p*-value = 0.8518) and the concentrations were similar between the two groups (*p*-value = 0.6889) (Figure 3). This demonstrates that the final libraries are stable and can be subjected to chemical inactivation and re-extraction without significant loss.

### 3.2. Preservation of Sample Quality Following Inactivation

Beyond the final libraries being of high enough quality for sequencing, we also wanted to determine if the inactivation and re-extraction of the final libraries would have an effect on the sample quality. To this end, we performed single-cell RNA sequencing on naïve mouse lymph nodes. The single-cell suspensions were negatively selected for myeloid cells to remove most lymphocytes. The libraries were either brought straight to sequencing after library prep or were inactivated according to the AVL protocol. All samples resulted in high-quality sequencing with Q30s for the barcodes, UMIs, and RNA all above 95%. The samples contained similar cell complexity (Figure 4A) and integrated wells (Figure 4B). Both samples had similar numbers of cells sequenced, similar amounts of RNA content per cell, and similar number of genes detected (Figure 4C,D). Together, this demonstrates that this method of inactivation is viable for the removal of samples from the BSL-4 laboratory without sacrificing the sample output and preserving sample complexity.

To further determine if there is a loss of any specific mRNAs, we performed differential expression analysis on the samples comparing the inactivated libraries to those treated normally. From this result, we found only 16 differentially expressed genes (adjusted *p*-value < 0.05 and fold change > 1). We performed enrichment analysis on these genes and found they were all associated with neutrophils and neutrophil signaling pathways (Table 1). Of the cell types identified, neutrophils had the highest degree of variability between all four samples (Figure 4A); thus, seeing gene differences associated with neutrophils is expected. This is not likely to be a result of the inactivation but more a feature of the cell captures since it is one specific cell type. Thus, we have shown that inactivation does not have an effect of differential expression results, but that sample pre-processing for cell counts can affect these results. Overall, the inactivation of the final library preserves the RNA content of the samples.

## 4. Discussion

The use of single-cell RNA technologies for the study of viruses requiring high and maximum containment comes with additional challenges. Here, we describe a method that is compatible with commonly used chemical reagents for the inactivation of nucleic acid material, which is applied to the final libraries obtained using a single-cell RNA sequencing method. Although demonstrated using the 10X genomics method for single-cell sequencing, this approach would be compatible with any single-cell sequencing kit or assay as the final libraries would all be short-fragment double-stranded DNA. Because of this, the proposed method would allow for laboratories to implement new single-cell technologies, such as Seq-Well, SPLiT-Seq, and PiP-Seq, without additional inactivation validation needed. In addition to being combatable with various platforms, it would also be compatible with a multitude of different sample collections such as CITE-Seq [21], ATAC-Seq [22], and ChIP-Seq [23] variations in the single-cell protocols. Finally, this method has already proven useful in the study of the host response to SARS-CoV-2 infections in non-human primates [11,13], demonstrating the high quality of data that is achieved after inactivation and removal from containment laboratories.

An additional benefit of taking samples to the final library stage is that when any new technologies arise or a company alters their reagents, there is no need to perform additional validation steps for inactivation. The final libraries are very stable and are even able to be irradiated, which is sometimes a valuable option when working with samples originating from positive sense RNA viruses. This stability also lends itself to additional inactivation chemicals being used as long as the double-stranded DNA library can be extracted on the other end.

A disadvantage to generating the full library inside the containment lab is the additional time required. However, when performing single-cell sequencing, the longest and most difficult steps will always include the moving of the samples through the initial barcoding stage. The additional steps of library prep are relatively short (Figure 2), often taking less than an hour, and there are multiple stopping points in every protocol that can allow for overnight incubations or time between the steps. In addition, in a companion article by Dr. Mühlberger and colleagues also published in *Pathogens*, the authors demonstrate how two consecutive heat steps after cDNA generation can inactivate the Nipah virus as a viable alternative to allow for an earlier exit from the BSL-4 [24]. However, this would need to be demonstrated at each lab individually and across all viral families, resulting in a high cost. An additional method for removing the samples included the Seq-well platform in [25] which immediately after the reverse transcription step, the sample was inactivated with the GeneXpert Lysis Buffer and then the beads were washed with the TE buffer.

## Figures and Tables

**Figure 1 pathogens-13-00167-f001:**
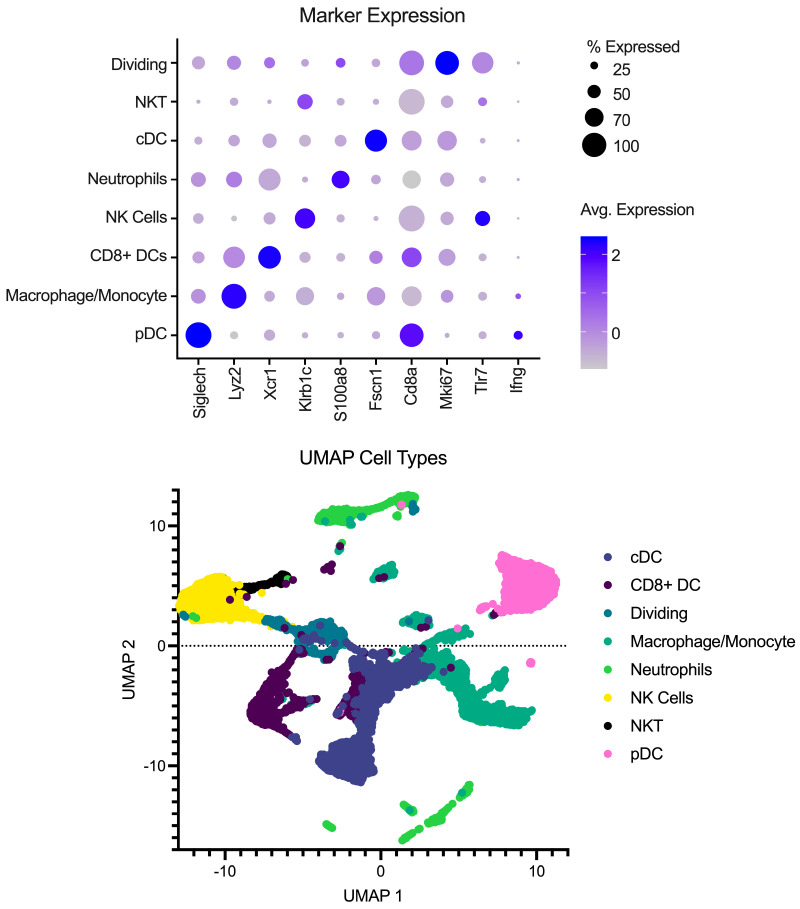
Overview of pre-processing of scRNA-Seq libraries. (**Top**) Marker expression of genes (columns) that identify the various cell types (rows). Average of normalized expression is shown in darker purple. (**Bottom**) UMAP projection with each point representing a different cell type colored by the cell annotation.

**Figure 2 pathogens-13-00167-f002:**
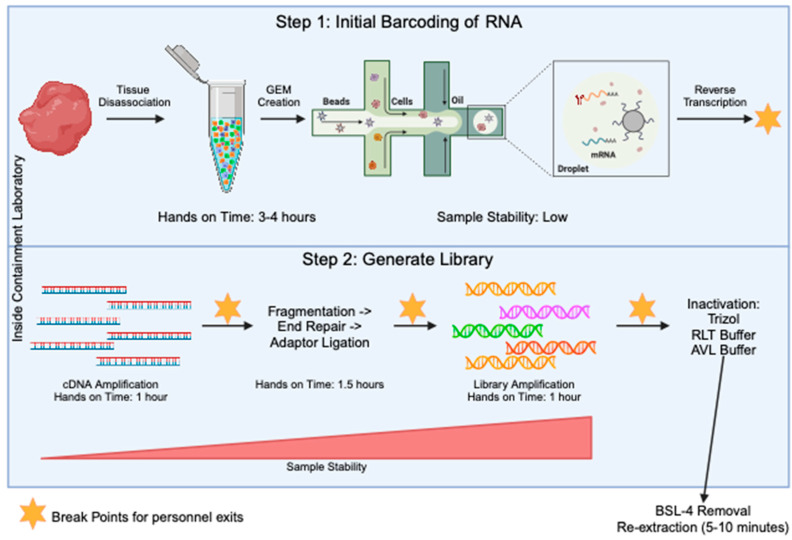
Overview of the 10X method for single-cell RNA seq library prep. The processes in step 1 (top blue box) are specific to the 10X Genomics protocol, but the processes in step 2 (bottom blue box) will be fairly conserved across all single-cell sequencing protocols. The steps in the blue boxes would all be carried out in the containment laboratory. The orange stars mark points in the protocol where there is a break and the sample can be stored for some time, allowing people to exit the lab. Sample stability and hands on time for each step is noted.

**Figure 3 pathogens-13-00167-f003:**
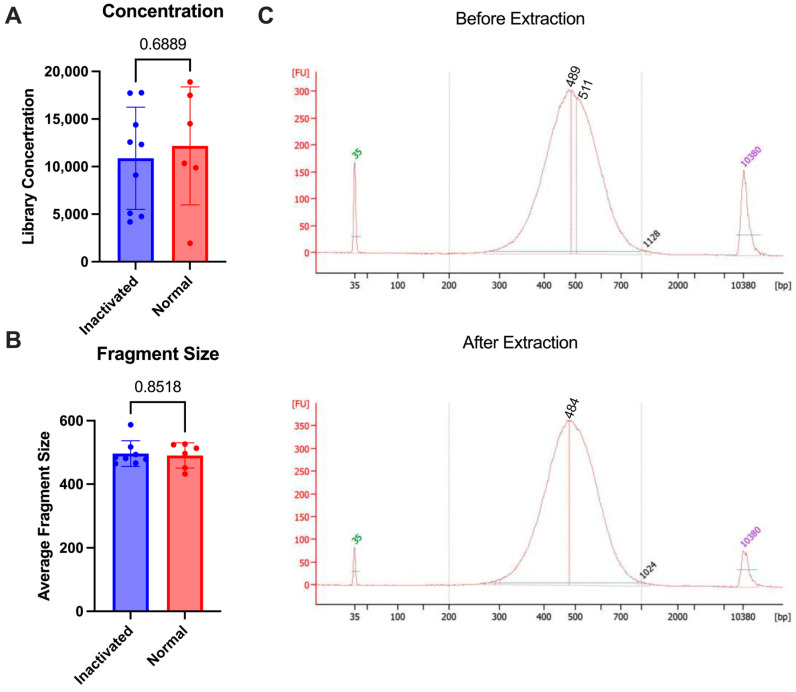
Comparison of 10X libraries with inactivation (blue) or without (normal, red). (**A**) Concentrations of libraries. (**B**) Average size of fragments. (**C**) Bioanalyzer trace comparison of a single sample before re-extraction (1:5 dilution) as compared with after (undiluted) using RLT buffer. *p*-values are determined by a Mann–Whitney test.

**Figure 4 pathogens-13-00167-f004:**
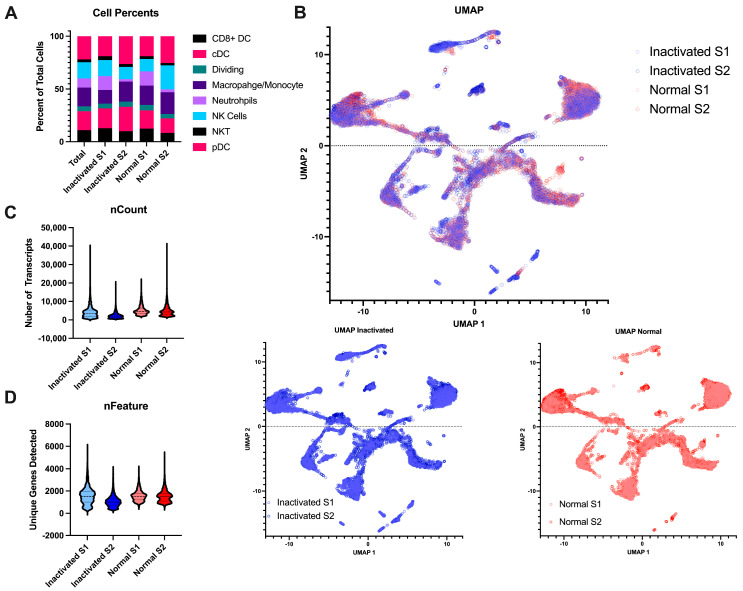
Comparison of inactivated samples to normal processed samples. (**A**) Stacked bar chart of the percent of the different myeloid cell subtypes detected per sample across the different conditions. (**B**) UMAP projections of the integrated data. The top is all the samples together with the inactivated samples in blue and the normal samples in red. The same projections with the type of post-library processing used can be observed below. (**C**) Violin plots of the number of unique mapped identifiers detected per cell in each sample. (**D**) Violin plot of the number of unique genes detected per cell across the 4 samples.

**Table 1 pathogens-13-00167-t001:** DE results of inactivated samples compared to normal samples.

	p_val	avg_log2FC	pct.1	pct.2	p_val_adj
Jchain	0	1.625376	0.266	0.528	0
S100a9	7.85 × 10^−141^	1.146031	0.281	0.25	1.57 × 10^−137^
Msrb1	9.21 × 10^−140^	1.057364	0.77	0.991	1.84 × 10^−136^
Cebpb	6.42 × 10^−110^	1.066698	0.775	0.981	1.28 × 10^−106^
Ccl21a	8.91 × 10^−97^	1.070559	0.239	0.314	1.78 × 10^−93^
S100a8	3.22 × 10^−94^	1.03576	0.289	0.294	6.44 × 10^−91^
Ptgs2	9.34 × 10^−74^	1.368172	0.448	0.573	1.87 × 10^−70^
Mzb1	4.30 × 10^−72^	1.290382	0.827	0.936	8.60 × 10^−69^
Slpi	3.91 × 10^−57^	1.379325	0.463	0.382	7.82 × 10^−54^
Cpa3	1.34 × 10^−54^	1.205746	0.286	0.36	2.67 × 10^−51^
Il1r2	1.58 × 10^−51^	1.16859	0.519	0.684	3.16 × 10^−48^
Dcpp1	1.75 × 10^−31^	1.194771	0.718	0.996	3.51 × 10^−28^
Clu	4.14 × 10^−26^	1.074544	0.329	0.356	8.27 × 10^−23^
Mcpt4	4.05 × 10^−12^	1.913468	0.396	0.424	8.09 × 10^−9^
Ifitm1	2.04 × 10^−11^	1.125462	0.394	0.338	4.07 × 10^−8^
Gsn	4.04 × 10^−10^	1.079454	0.771	0.66	8.08 × 10^−7^
Cma1	1.32 × 10^−7^	1.404933	0.443	0.42	2.65 × 10^−4^
Il1b	4.18 × 10^−7^	1.562076	0.552	0.593	8.36 × 10^−4^
Cxcl2	5.47 × 10^−6^	2.385164	0.329	0.384	1.09 × 10^−2^

## Data Availability

The data presented in the article can be acquired upon request of the R object for the sequencing samples.

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
