# Peer review of "Preservation of scRNA-Seq Libraries Using Existing Inactivation Protocols"

_pathogens, 2024, doi:10.3390/pathogens13020167_

Round 1

Reviewer 1 Report

Comments and Suggestions for Authors

Single cell sequencing technologies are applied to study of infectious diseases widely. With the power to deeply profile individual cellular responses to infection we can now generate a deeper understanding of how the host response to infections is coordinated. But the big challenger is that most of experiments need be done in high containment laboratories under the biosafety level 3 and biosafety level 4 (BSL-4) conditions. Library requires extensive testing for inactivation of pathogens before the sample can be sent out for sequencing. The authors tested a couple reagents and protocols for sample inactivation such as Trizol, RLT, and AVL buffers and demonstrated that these simple methods of sample prep preserve the sample quality allowing for downstream analysis. The kind of study is valuable.

 Some questions/issues need be answered:

1.      Provide summary of sequencing quality data for each sample. Inactivation reagents may degrade mRNA/cDNA. We must check the sequencing quality.

2.      For scRNA-seq result, you should provide a). UMAP plot labeled with cell types; 2) feature plot or dotplot of marker genes for each cluster; 3) a table of cell number or percent of  total cell number.

3.      Call differential expression genes between normal and inactivated samples; Then do GSEA(pathway or GO) analysis for DEGs which can tell us if any RNA lost during inactivation.

Author Response

We would like to thank the reviewer for the kind works. We have provided some additional information as was requested by the reviewer. Please see our responses below.

 Some questions/issues need be answered:

  1. Provide summary of sequencing quality data for each sample. Inactivation reagents may degrade mRNA/cDNA. We must check the sequencing quality.
    1. In the manuscript, we are describing inactivation only at the stage of final library to avoid any potential degradation of mRNA/cDNA as these are more fragile stages of the prep. In Figure 2 we show that before and after extraction, there is no significant alteration in the library concertation’s or fragment sizes as can be seen by the bar graphs and the example bioanalyzer trace. This signifies that the libraries are still of good enough quality for sequencing. We have added that for Figure 3, all data contained high PHRED scores from sequencing and had Q30 > 95% for the barcodes and for the inserts. Additionally, we show that there is not a loss of complexity after extraction in figure 3 demonstrating that the same genes were sequenced even with the inactivation steps (same amount of unique transcripts). Together we think this supports that the inactivation used does not significantly alter the results. Finally, in the methods we have added an additional figure to show the pre-processing that went into the single cell analysis (see below)
  2. For scRNA-seq result, you should provide a). UMAP plot labeled with cell types; 2) feature plot or dotplot of marker genes for each cluster; 3) a table of cell number or percent of total cell number.
    1. For this study, the focus was not on the single cell sequencing cell annotations. However, as requested, we have provided in the methods information on the cell annotations including a representative marker gene for each cell type. Figure 3A is a table of the percent of total cell number across the different samples. We have added a column in this figure for a comparison to the total aggerate samples.
  3. Call differential expression genes between normal and inactivated samples; Thendo GSEA(pathway or GO) analysis for DEGs which can tell us if any RNA lost during inactivation.
    1. We have provided DE analysis in the single-cell RNA seq data and form the global dataset only found 19 genes that were differentially expressed between inactivated samples and samples that were treated normally. These genes were associated with neutrophils. This is not surprising as neutrophils had the most unstable cell counts across the different samples contributing to the changes in their gene expression. This is expected as neutrophils are difficult to capture during the 10X processing and samples have undergone extensive pre-processing before being separated on the 10X genomics chips. Thus, we think this is likely due to sample handling and not an effect form the inactivation which did not occur until well after the RNA had been captured, barcoded, amplified, fragmented, ligated, and amplified again. There were no other significant changes in gene expression. We have added this information to the text.

Reviewer 2 Report

Comments and Suggestions for Authors

In this article authors presented a method for performing single-cell sequencing from samples generated in  high containment laboratories and outlined a method for sample inactivation of infected samples (infectious) using known reagents and SOPs widely used sample inactivation. Their work also is beneficial to wide group of scientists saving scientific resources, time and enhancing the research technologies. I have some minor comments regarding the presentation of this elegant piece of work and some reorganization.

An additional benefit of taking samples to the final library stage is that when any 204 new technologies arises or a company alters their reagents, there is no need to perform 205 additional validation steps for inactivation.

Minor Comments

1.     I missed the A549 cells in results.

2.     Authors can use the term Activated in place of Normal when comparing to Inactivated group Fig 2 and 3

3.     Fig 1-3 legends are describing part of results. Authors can represent all figure legends as more informative. Succinct description of each step may help the readers. “The step 1 is specific to the 10X Genomics protocol, but step 2 will be fairly conserved across all single cell sequencing protocols.’ 

4.     Fonts and text size look different in figures.

5.     Statistics section is unclear.

Comments on the Quality of English Language

NA

Author Response

We would like to thank the review for their thoughtful comments. We agree this could be very useful information for individuals wanting to perform single cell studies in high containment. Below are out responses to the reviewer comments.

  1. I missed the A549 cells in results.
    1. The A549 cell results are included in Figure 2 (now figure 3). That figure represented an aggregation of multiple studies to show the inactivation does not affect the final library. We did not perform sequencing on the A549 libraries, only on the mouse lymph nodes. We have clarified this point in the text.
  2. Authors can use the term Activated in place of Normal when comparing to Inactivated group Fig 2 and 3.
    1. Though we do see the reviewers point here, we prefer not to use the term Activated as that is not always the case. They are simply samples that were processed normally and without the inactivation steps being carried out on the final library. The term Activated could imply that we had “active” infected samples that were sequenced, which is not correct and we would not like for someone to think we were handling those samples in an unsafe way. Though no live agent was used in this study, we think the term activated would imply it was and could create problems.
  3. Fig 1-3 legends are describing part of results. Authors can represent all figure legends as more informative. Succinct description of each step may help the readers. “The step 1 is specific to the 10X Genomics protocol, but step 2 will be fairly conserved across all single cell sequencing protocols.’ 
    1. We have clarified the language in the figure legends a bit and hope this helps.
  4. Fonts and text size look different in figures.
    1. We have tried to make the fonts more consistent across the figures. Thank you for noticing the issues.
  5. Statistics section is unclear.
    1. We have added clarity to the statistics section.